# Thriving, Not Just Surviving in Changing Times: How Sustainability, Agility and Digitalization Intertwine with Organizational Resilience

**Antonio Miceli** [1], **Birgit Hagen** [1], **Maria Pia Riccardi** [1,2], **Francesco Sotti** [1] **and Davide Settembre-Blundo** [1,3,*]

1 Department of Economics and Management, University of Pavia, Via San Felice 7, 27100 Pavia, Italy; antonio.miceli01@universitadipavia.it (A.M.); birgit.hagen@unipv.it (B.H.); mariapia.riccardi@unipv.it (M.P.R.); francesco.sotti@unipv.it (F.S.)

2 Department of Earth and Environmental Sciences, University of Pavia, Via Ferrata 9, 27100 Pavia, Italy

3 Gruppo Ceramiche Gresmalt, Via Mosca 4, 41049 Sassuolo, Italy

* Correspondence: davide.settembre@gresmalt.it

**Abstract:** Nowadays, the buzzwords for organizations to be prepared for the competitive environment's challenges are sustainability, digitalization, resilience and agility. However, despite the fact that these concepts have come into common use at the level of both scholars and practitioners, the nature of the relation between sustainability and resilience has not yet been sufficiently clarified. Above all, there is still no evidence of what factors determine greater resilience to change in an organization that also wants to be more sustainable, especially in times of crisis and discontinuity. This research aims to explore from a theoretical point of view, through the construction of a conceptual model, how these dimensions interact to help the business to become strategically resilient by leveraging digitization and agility as enablers. A new view of resilience arises from the study, which goes beyond the well-known ability to absorb or adapt to adversity, to also include a strategic attribute that could help companies capture change-related opportunities to design new ways of doing business under stress. A key set of strategically agile processes, enabled by digitalization, creates strategic resilience that also includes a proactive, opportunity-focused attitude in the face of change. Strategic resilience to lead to organizational sustainability must be understood as a multi-domain concept quite similar to the holistic view of sustainability: environment, economy and society. Finally, the research offers a set of propositions and a theoretical framework that can be empirically validated.

**Keywords:** strategic resilience; multi-domain resilience; sustainability; strategic agility; digitalization; crisis; change

## 1. Introduction

Organizations to better adapt to change, they must some essential attributes, and nowadays the buzzwords become: sustainability, digitization, resilience and agility. In today's dynamic and interconnected world, organizations must be able to cope and thrive in conditions of crisis and change. Change can be permanent or discontinuous, incremental or radical, be expected or unexpected, reversible or irreversible and can come with varying intensity and at many different levels. This highlights the degree of complexity and multidimensionality characteristics of change [1] that influence the development and implementation of effective business strategies and solutions. In addition to occasional disruptions and continuous incremental changes, radical changes, discontinuities, crises or "crashes" are becoming increasingly important and frequent, all of which have a great impact on the operations of organizations [2]. The COVID-19 pandemic is only the most recent and dramatic example of a global crisis that highlighted, in a relatively short time, the far-reaching implications and challenges of change as well as the urgency of efficient and effective responses [3].

The crisis that developed globally following the spread of the COVID-19 pandemic has strongly highlighted the importance of sustainability, not only in the health aspect of its social dimension, but also from an economic and environmental point of view [4]. In fact, the health emergency immediately produced a crisis both on the offer side (blockage of supply chains, closure of factories, significant drop in employment) and on the demand side with the collapse of household consumption and business investments [5]. The growth of economies was, therefore, abruptly interrupted by the pandemic, also in the perspective of sustainable development as defined by Agenda 2030, putting an end to years of efforts by countries and the international community to achieve progress in the level of well-being of their populations [6]. It is, therefore, clear that future growth and development policies cannot ignore the ability of societies, organizations and individuals to face the complex and unpredictable risks and phenomena that continually afflict them. This capacity is commonly known as resilience, and the health crisis linked to COVID-19 has shown how it correlates with environmental and socio-economic sustainability. The nature and modalities of correlation between resilience and sustainability have not yet been appropriately investigated [7,8].

Companies and organizations have also shown that they are not prepared for a global health emergency represented by the COVID-19 coronavirus pandemic [9]. This criticality has highlighted the need to develop a rapid capacity to adapt to change, quickly implementing effective actions to respond to the current situation [10]. In particular, companies need to be prepared to face the potential for prolonged impacts on operations, supply chains and the economy in general and, of course, on staff welfare resulting from a globally spreading infectious disease. Companies should take advantage of this critical moment to review their strategies and plans in response to the crisis to be well prepared for the potential impacts of the pandemic by increasing innovation capacity in a way that firms should become more resilient [11].

The term resilience has been widely used with the outbreak of the coronavirus emergency. Among the most recurrent statements is that resilient economies are the ones that will emerge best from the crisis, or that resilience is needed to restart. While governments and institutions talk about resilience with a view to building or maintaining it, it is not always clear what the exact meaning of this term is [12]. The term itself indicates the ability of a system to withstand any disturbance, organizing a response and returning to normal operation. However, resilience is not to be confused with resistance to change, since partial transformation of the parts is allowed. This ability stems mainly from the intrinsic characteristics of each system, which allow us to effectively overcome critical situations [13]. More specifically for an organization, elements such as knowledge, the ability to react to change, openness, availability of adequate resources, flexibility and a wide network of relationships allow an organization to be resilient [14].

Although resilience is a very topical concept that has received increasing attention from scholars over the past two decades in various research domains [15], nevertheless it has been conceptualized theoretically quite heterogeneously, resulting in the proliferation of different definitions, approaches, theories and interpretations [16]. The orientation of researchers has primarily focused on the conceptual meaning of resilience as an "absorptive" or "adaptive" capacity by borrowing definitions commonly used in systemic and evolutionary theory [17]. What is lacking in the body of research, however, is a theoretical discussion of the factors that make organizations more resilient, thus, able to successfully cope with change [18]. Likewise, there is a lack of theoretical exploration regarding how resilient organizations can prepare for and respond to change in a sustainable way [19].

Based on the foregoing, this research seeks to investigate the elements that identify, at least potentially, the resilience of an organization [20] also by relating it to other distinctive capabilities or enablers such as technological innovation or digitalization [21]. Secondly, it aims to fill conceptual gaps with describing the relationship between resilience and sustainability by designing a framework that also explores and maps ways to build them in the organization.

## 2. Methodology and Research Aim

An exploratory conceptual research in order to fill the theoretical gaps highlighted in the previous section is the methodology followed in this study. This approach has already been widely applied in theoretical research in the organizational field [22,23]. Through a critical review of the literature, the existing knowledge of the relationship between resilience and sustainability in management practice was identified in order to design a conceptual model. The literature analysis was carried out through a bibliographic research of articles written in English and published in the Scopus repository and Google Scholar database. The keywords and terms used for the search were: "resilience", "sustainability", "digitalization" and "agility", combined with an "AND" and/or an "OR" [24].

The review of the literature thus has been conducted with the critical review approach, an appropriate method to address new topics in order to allow new interpretations and perspectives [25]. This approach differs from the systematic literature review, because it does not claim to include all published articles, but rather to combine viewpoints and insights from different fields of research to generate a novel conceptual framework or theory. Finally, the organizational framework of the critical review has been determined through the enunciation of constructs that help to make sense of the accumulated knowledge on the topics [26,27]. Below is the review of the literature organized into three sub-sections, which bring together the main concepts of the topics identified by the keywords.

## 3. Towards a Conceptual Framework

Two excellent and very recent systematic literature reviews [16,17] facilitated taking stock of the work on resilience in management, the discussion of resilience in sustainability literature instead helped in identifying and challenging some of the underlying assumptions [28,29].

### 3.1. Clarifying the Concepts of Resilience and Sustainability

#### 3.1.1. Resilience

Resilience is an increasingly popular concept in both management practice and in scholarly research [30], encompassing not only the complexity of organizations, but also other contexts, for example, urban contexts [31,32]. It deals mainly with reactions to adversity and reflects the growing complexity and interdependence of socio-economic, financial and technological systems, the associated challenges for businesses and the need for solutions to deal with unexpected or unpredictable change [23]. According to [16,17,33,34], academic inquiry has been undertaken at organizational level as a response to external threats, employee-level resilience and strengths, but also at business models or supply chain levels in order to explore adaptability or design principles that reduce vulnerabilities and disruptions. Thus, setting the boundaries is a formidable challenge, because of the many forces and types of change, the interdependencies and the levels of systems that come into play when discussing resilience.

According to the reviews [16,17], there are two dominant interpretations of resilience. One view relates to "absorption" or the system's ability to "bounce back". In this case, there is no real adaptation (i.e., development with change) to changing conditions. Such a "backup" system provides the same functionality to sustain the system exactly as it was before the change. Adaption instead, the second interpretation, includes a capacity to modify incrementally its functions in the face of change. In this way, the system "learns" and evolves with change [35]. Following this distinction, the understanding of the magnitude of "disturbance" that the system can tolerate [36] is important, as it will determine the response and the route of action in the face of change [37]. According to [38] and [39], it is essential not only to understand the nature of change, but also the system's degree of self-organization and the system's learning and adaptation capacities need to be understood.

What also remains to be discussed and clarified is whether the initial conditions and/or the new conditions are the (more) favorable conditions. In the first case, i.e.,

absorption, resilience work assumes that the initial conditions are more favorable than are the new ones—the aim is to bounce back; in the second case, adaptation, work seems to assume that the new conditions are more favorable than the initial ones. Under the same line of thought, we may add that neither absorption nor incremental and continuous adaptation may be an adequate response to changed conditions. With a view to thrive more than to survive, the organization may not only need but want to actively shape conditions, seek to prepare for and address change for long term growth and aim at transformation and renewal.

### 3.1.2. Sustainability

Sustainability and sustainable development are two terms that are mentioned more and more often and are linked to ever wider areas. Despite the great attentiveness, there is still a tendency to attribute different and sometimes discordant meanings to this term [40]. The generic definition of sustainability can be traced back to the 1980s with the drafting of the Brundtland Report by the World Commission on Environment and Development (WCED). In that document, sustainability was identified with the satisfaction of the needs of the present generation without compromising the ability of future generations to meet their own needs, ensuring a balance between economic growth, environmental protection and social well-being [41]. This is also the origin of the idea of sustainable development, as a way of progress that maintains this delicate balance today, without endangering the resources of tomorrow. It, therefore, refers to an approach that calls for the adoption of development strategies that take into account both the observable short-term effects (sustainability) and the long-term effects (sustainable development) [42]. A number of other concepts have since emerged from this foundation, such as the definition of environmental sustainability, which is one that emphasizes the preservation of biodiversity without sacrificing economic and social progress [43]; economic sustainability, which ensures that activities that seek environmental and social sustainability are profitable [44]; social sustainability, which seeks population cohesion and stability [45]. Thus, sustainability and sustainable development work on the principle that available resources cannot be used indiscriminately, that natural resources must be protected, and that all people must have access to the same opportunities.

Sustainability and sustainable development have become a central issue for companies, regardless of their size or industry [46]. Companies are called to seek a balance between economic, social and environmental benefits [47], an objective which is well illustrated with the call for the triple P (people, profit and planet) bottom line, which defines sustainability as the intersection of environmental, social and economic value [48,49]. Environmental value can be achieved through the use of renewable resources and the reduction in waste and emissions [50]; the social value derives from the social development and the well-being generated by the organization [51]; while the economic value commonly is seen in terms of survival, growth or long-term performance of the firm. Frequently, economic value is seen as a consequence of organizational resilience [52].

### 3.1.3. The Link between Resilience and Sustainability

In times of uncertainty, commitment to sustainability is essential for companies that also want to be resilient [53]. The two concepts, although different, share the same goal: to achieve sustainable development [54]. Accompanying organizations and society as a whole towards a state of equilibrium that must be maintained as stable as possible is a challenging task [55]. A resilient system is one that has the ability to resist and recover from impacts and disruptions [56]. While a resilient development is one that adapts to changing conditions and can recover from extreme and adverse circumstances [57]. Therefore, the resilience of organizations helps to deal with the complexity of change, while preserving the capacity for development. The governance of these organizations in such complex scenarios should focus on achieving organizational resilience based on sustainability (economic, social and environmental) that ensures the achievement of their sustainable development goals [58].

The pandemic crisis of the COVID-19 has changed the paradigm of business performance evaluation [59]. While in the past the entrepreneur was responsible to the investor for the company's ability to generate profits [60], now all stakeholders (and not just shareholders) also want to know how those profits were generated. Therefore, they want to know if the environment (environment), the rights of workers and more generally the community (social) and the basic rules of corporate governance (governance) have been respected [61]. By now, we are consolidating the awareness that the value creation of companies is closely related to environmental, social and governance (ESG) performance even in times of crisis [62,63]. As a result, companies that are now more attentive to ESG factors, which in the past was only the prerogative of large listed companies, are also more resilient to risks produced by complex and unpredictable phenomena such as health emergencies. When talking about resilience and sustainability, it is important to remind us that there is a strong relationship between these two concepts, as they share many features [64] As a matter of fact, resilience and sustainability have in common many research methodologies, since both concepts are focused on the topic of system survivability [65]. According to the framework designed by Marchese [64], resilience can be considered as a component of the wider concept of sustainability; in fact, this point of view relies on a notion that assumes that an increase in the system's resilience leads to an increase in the system's sustainability [66]. The integration of sustainability and resilience allows to generate a comprehensive system approach, which is crucial to establish a sustainable decision making based on the consideration of industrial, social and ecological dynamics [67]. When discussing the ecological–economic system, it is important to consider that resilience is one of the elements that must be considered when planning the organization's strategies for sustainable management [68]. The connection between resilience and sustainability leads to the establishment of a new approach known as "resilient sustainability", which represents a driving mindset, allowing decision makers to find solutions to sustainability issues by leveraging on the adaptative capacities of the company in order to find the best strategies [69].

If sustainability is defined holistically, the required balance and the interplay of the three values (environmental, social, economic) necessitates a discussion of whether and how resilience can contribute to overall organizational sustainability, or, more in general, whether and how the two concepts are related [70]. As mentioned above, the dominant interpretations of resilience in the management context prescribe absorption, bouncing back to initial conditions, or adaptation towards reaching a new equilibrium without an explicit consideration of whether the initial conditions or the new conditions are more favorable or "sustainable" [71]. Thus, a system may be resilient but unsustainble [72]. Additional complexity is added by the fact that there may be a trade-off across the three values and that a balance is required. It follows that resilience, similarly to sustainability, must be broken down in sub-domains, since resilience in one system (economic, ecological, social) does not automatically lead to a positive impact in another system. An in-depth discussion of resilience - in terms of resilience of "what" to "what"- therefore, would help us to understand and qualify interconnections and responses better.

On the other hand, sustainability may be thought of as an element to build resilience. Companies whose practices and strategies include considerations of environmental, social and economic balance may also be more resilient to change [53]. For example, production processes or supply chains with lower environmental and social impact may immunize against environmental jolts or social upheaval. They may also bring benefit in terms of market positioning and, thus, contribute to resilience and value creation [73].

An in-depth discussion of the interdependencies and the causality between the broad view of sustainability and resilience goes beyond the scope of this work. Here, we see sustainability holistically, as a metaobjective and a basis for (normative) performance indicators of the resilient organization. Consequently, we postulate that:

**Proposition 1 (P1).** *Sustainability and resilience can be viewed as two interdependent concepts.*

**P1a.** *It remains to be clarified how resilience affects sustainability (understood in a holistic sense) and vice versa.*

**P1b.** *Resilience may induce positive outcomes in one of the sustainability domains (ecological, economic, social—EES), but it may not automatically induce the same positive effects in the other ones.*

**P1c.** *Sustainability may improve resilience, but the positive outcome experienced in one resilience domain does not automatically lead to a positive outcome in another one.*

**Proposition 2 (P2).** *If sustainability is understood as a multi-domain concept and ultimate meta objective of the organization, resilience should also be defined correspondingly to include economic, environmental and social resilience.*

**P2a.** *Establishing a multi-domain concept of resilience allows us to account for interconnections and trade-offs/synergies with regard to multi-domain sustainability. This implies that resilience must include a normative element regarding the desirable outcome and/or desirability of system conditions.*

**P2b.** *Resilience in one domain does not automatically lead to resilience in another domain.*

**P2c.** *Sustainability in one domain does not automatically lead to sustainability in another domain.*

### 3.2. Building the Strategically Resilient–Agile Organization

According to [29], research so far has attempted to make sense of events in a given period to generate insights into how organizations (should) deal with adversity under a particular set of circumstances. Insights from the various research streams essentially point to the importance of slack and, thus, call for the accumulation of resources to build redundancy and resourcefulness (i.e., variety). The approach is in line with the ideas of evolutionary theory, which also builds the theoretical background for the conceptualization of resilience as adaptation. Additionally, the "structure" of the firm and its processes, for example in terms of losing control or flexibility are seen to promote resilience. These findings link to the supply chain work and emphasize facilitated resource reconfiguration [74], which is seen crucial to respond to changing conditions. Finally, a wide range of enablers in terms of information processing and communication, collaboration and networking is mentioned. Generally, the strategy of accumulating resources has been criticized not only for being inefficient and costly but also difficult in terms of "reconfiguration" for different purposes [75].

Overall, resilience research is yet to identify and understand the factors that build organizational resilience to future conditions [76]. Hamel and Välikangas [77] draw on insights from innovation research to propose that to become strategically resilient, the organization must address a cognitive, a strategic, a political and an ideological challenge. Their idea of resilience relates to a broad, strategic view of resilience in terms of an organization's renewal and transformation. The cognitive challenge relates to the idea that a company must build deep awareness of what is changing and be willing to consider how those changes will affect its current success. The strategic challenge consists of developing a range of new options that represent alternatives to dying strategies, while the political challenge relates to resource re-allocation to future programs. Finally, the ideology to be challenged is the quest for operational excellence and flawless execution. The approach proposed by [78] (i.e., to design strategically agile business processes to thrive in conditions of uncertainty and change) nicely fits with this line of thought, as it helps cope with these challenges and, thus, can be considered a driver of strategic resilience. The objective to thrive includes absorption, adaptation but also transformation and renewal to actively prepare for and achieve growth on the longer term. Importantly, it includes the notion of speed, which is, surprisingly, neglected in most management resilience work with the notable exception of the supply chain research strand [79,80]. The question of how quickly the organization can absorb, adapt or transform and renew is of huge importance, especially in a dynamically and quickly changing business landscape.

In [78]'s view, designing flexible and responsive processes for (1) information search, (2) business development or innovation, (3) harmonization and coordination of the value chain and (4) resource mobilization and leverage make the firm strategically agile. According to the authors, such agile processes help the firm to move in conditions of unpredictability for which discontinuities or the consequence of change make a case. For example, in a situation where information is unavailable or contradictory, the authors propose to base information searches on close and regular contacts with a diverse range of customers and partners in order to help the organization read signals early, interpret them better and act on them efficiently and effectively. Customers and partners should also make part of the business development/innovation process, which is based on experimentation. The idea of experimentation for innovation is not new, but the idea to opt for multiple small experiments to develop strategic agility or resilience is [81]. In our context, such a way of approaching innovation has two major advantages: it prepares a variety of options in advance, while limiting the risk of "big" failure [82]. It, therefore, optimizes resource allocation instead of a potentially inefficient and costly accumulation of resources to guarantee slack. It is, thus, a flexible and strategic way of developing a variety of options, and developing them early, so that they can be put into use once the context is changing. Customers and partners should take part to the process so that resources and risk are shared, and the innovation is responsive to their needs [83]. At the same time, the innovation is tested and adapted early and quickly, reducing which again reduces the risk of lost investment while at the same time accelerating the innovation process and the response.

The third process, the coordination and harmonization of the value chain, is based on the idea of flexibility and responsiveness in terms of resource generation. Partners may compensate for the organization's lack of resources or contribute to their variety. Relationships, furthermore, may be more or less actively used under different conditions [84]. Importantly, the view of an internally and externally harmonized and coordinated value chain includes the notion that the various partners do act with the same "strategic agility" philosophy, so as to achieve coordinated and timely action, accommodating a "system view" and accounting for interdependence [85]. Finally, the process of resource mobilization links to the value chain and the idea that resources do not need to be developed and owned by the organization itself but can be mobilized externally or be co-developed and owned. Secondly, a creative use of resources, leveraging the resources at hand, involves creating synergies, for example regarding data collection, analysis and use, co-promotion, co-development of offerings, etc. [86]. Overall, designing and implementing processes with a view on strategic agility will guarantee a resource-conservative way to cope with the need for absorption, adaptation and renewal. At the same time, it ensures "preparedness" and a timely and effective response to unexpected change or, more in general, situations of uncertainty.

As is clear from the above, the concepts of strategic resilience and strategic agility are similar. What they have in common and what differentiates the concept from resilience, as it is predominantly discussed in management literature, is the view on change as opportunity and active, timely management of change towards renewal and long-term growth—as compared to survival and dealing successfully with adversity [87]. Finally, while the speed of response has received little attention in the overall body of resilience work, it is emphasized with the concept of agility. The dimension is also reinforced in the context of supply chains. Shekarian et al., for example [79], posit that above all the agile supply chain is fast in perceiving and responding to changing market needs and at the same time able to include the opportunity to improve delivery systems.

When resilience and agility are compared instead, again, the supply chain literature, which has discussed both concepts, may inform our work. Recently [88], thanks to an exhaustive analysis of the literature, have clarified the two concepts of agility and resilience in an integrated way by structuring differentiating and common features. In Figure 1, these dimensions are illustrated.

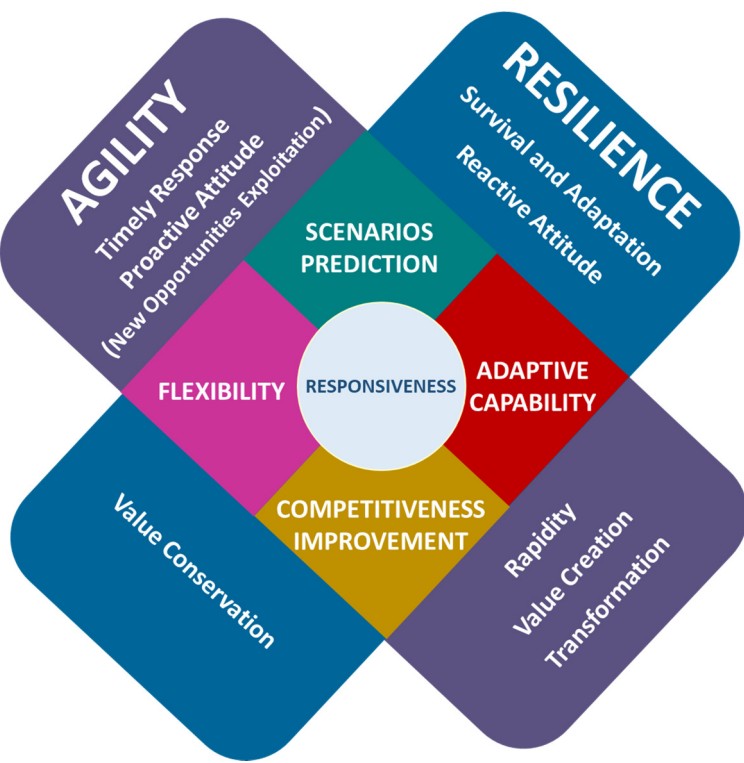

**Figure 1.** Distinctive and common features of resilience and agility (adapted from [88] and [89]).

The focus here, however, is not on discussing commonalities and differences but on the ways forward to combine the concepts in a way that allows an organization to build resilience strategically. Based on the above we propose.

**Proposition 3 (P3).** *Agility builds a strategic dimension of resilience, i.e., the capability of an organization to manage change proactively, more effectively and efficiently with a view to transformation and renewal. It includes the notion of the speed of the organization's response to change.*

### 3.3. Digitalization

Simply put, business digitalization consists of the implementation of digital tools and technologies as well as data, which together can make business processes more efficient [90] and effective. It is this aspect of digitalization that is of particular interest to our discussion of how to build strategically agile processes and, in turn, strategic resilience and sustainability. Digital technologies allow the collection of huge amounts of data, which are constantly increasing not only in quantity, but also in diversity [91]. Big data analytics for predictive analysis recognizes patterns that signal upcoming events and identifies measures to solve issues and improve outcomes [92,93]. Under the same line of thought, the contribution of artificial intelligence (AI) will be fundamental to analyze business data and provide a system that makes "complex thinking" easier and may build an improved basis for decision making [94] assisting both the identification of change as well as its management.

Digitalization also helps through an unprecedented potential of interconnection of business processes [95] and of stakeholders. Ensuing is the ability to better monitor activities to improve organization and coordination and quality of work [96] and to adapt more quickly to changing market conditions [97]. The basis of this "model" is the availability of all relevant information in real time that connects, through digital technologies, all stakeholders involved in the value chain, internally and externally. Digitalization also allows us to keep significant business processes operational when unexpected events occur, minimizing the economic impact and safeguarding the functioning of (production)

processes in the medium and long term [98,99]. Digital technologies that enable remote collaboration, virtual process management and real-time connectivity provide the means to respond effectively and efficiently [100,101].

In summary, digitalization has the potential, through processes, to mitigate the magnitude and reach of change by, at the same time, increasing the proactive stance and agility of business processes and the resilience of the organization. Based on this, we can argue:

**Proposition 4 (P4).** *Digitalization, through data and technologies, promotes agility because it increases the flexibility and responsiveness of the organization's business processes, for example by identifying changes early and by enabling efficient and effective connection and coordination of business processes and partners.*

Digitalization is also improving the sustainability of companies, enabling them to produce in a more environmentally friendly way [102]. Digital technologies increase the operational efficiency through the accessibility and collection of process data in real time, the management of energy and resource consumption and knowledge of the entire life cycle (design, manufacturing, distribution, maintenance and use) with the potential to eliminate discontinuities and inefficiencies [103]. The connection of processes and products, value chain and users allows the design of the product's manufacturing cycle together with that of its use in a logic of environmental and economic sustainability. In this way, it is possible to optimize the consumption of resources and reduce energy inefficiency and waste generated along the entire value chain [104]. Thereby, we can state:

**Proposition 5 (P5).** *Digitalization, by itself, enables efficient use of resources, thus contributing positively to sustainability.*

However, technology, including digitalization, is not neutral, especially when assessed from a socio–environmental–economic systems perspective [105]. Rebound effects and "turbulence", for example technological or industry disruption generated or triggered by digitalization, therefore, must be assessed. The ongoing discussion related to challenges and tensions around the Industry 4.0 illustrate the complexity and interdependence of the systems [106]. Therefore, we propose:

**Proposition 6 (P6).** *Digitalization, by itself, triggers change and rebound effects may occur. Thus, it may have also a negative influence on:*

**P6a.** *Resilience and on*

**P6b.** *Sustainability.*

In conclusion, we add the last proposition to the discussion on how to become a strategically resilient organization. One may propose that having built-in strategic resilience in the organization's processes and systems, this competence will be leveraged continually in the face of change. Under this line of thought, the strategically resilient organization establishes a virtuous cycle with regard to the management of change [107]. Thus, the following is being proposed.

**Proposition 7 (P7).** *The strategically resilient organization continually leverages and strengthens its competences in change management to establish a virtuous circle.*

## 4. Designing a Conceptual Model

This paragraph illustrates the design process of the conceptual model that aims to identify the characterizing elements and to formalize the links between them. The pivotal elements represented by the model are:

- The constructs that are the attributes of reality built through theoretical reasoning;

- The propositions that are the concepts that relate the different constructs.

The Table 1 offers an overview of the propositions that illustrate the interplay between the key concepts of the conceptual framework.

**Table 1.** Propositions overview (own elaboration).

| | Proposition | Construct 1 | Construct 2 |
|---|---|---|---|
| P1 | Sustainability and resilience can be viewed as two interdependent concepts. | Sustainability | Resilience |
| P1a | *It still remains to be clarified how resilience affects sustainabilty and vice versa (understood in a holistic sense) and vice versa.* | | |
| P1b | *Resilience may induce positive outcomes in one of the sustainability domains (ecological, economic, social—EES), but it may not automatically induce the same positive effects in the other ones.* | | |
| P1c | *Sustainability may improve resilience, but the positive outcome experienced in one resilience domain does not automatically lead to a positive outcome in another one.* | | |
| P2 | If sustainability is understood as a multi-domain concept and ultimate meta objective of the organization, resilience should also be defined correspondingly to include economic, environmental and social resilience. | Resilience | Sustainability |
| P2a | *Establishing a multi-domain concept of resilience allows us to account for interconnections and trade-offs/synergies with regard to multi-domain sustainability. This implies that resilience must include a normative element regarding the desirable outcome and/or desirability of system conditions.* | | |
| P2b | *Resilience in one domain does not automatically lead to resilience in another domain.* | | |
| P2c | *Sustainability in one domain does not automatically lead to sustainability in another domain.* | | |
| P3 | Agility builds a strategic dimension of resilience, i.e., the capability of an organization to manage change proactively, more effectively and efficiently with a view to transformation and renewal. It includes the notion of the speed of the organization's response to change. | Agility | Resilience |
| P4 | Digitalization, through data and technologies, promotes agility because it increases the flexibility and responsiveness of the organization's business processes, for example by identifying changes early, and by enabling efficient and effective connection and coordination of business processes and partners. | Digitalization | Agility |
| P5 | Digitalization, by itself, enables efficient use of resources, thus contributing positively to sustainability. | Digitalization | Sustainability |
| P6 | Digitalization, by itself, triggers change and rebound effects may occur. Thus, it may also have a negative influence on | Digitalization | Resilience Sustainability |
| P6a | *resilience and on* | | |
| P6b | *sustainability.* | | |
| P7 | The strategically resilient organization continually leverages and strengthens its competences in change management to establish a virtuous circle. | Resilience | Change Management |

As it is possible to notice from the table, seven propositions have been formulated. The first one (P1) highlights the interdependence of the concepts of sustainability and resilience by pointing out that the effects of one on the other, and vice versa, can be both positive and negative. The second proposition (P2) calls for a multi-domain view

of resilience, just as is the case for sustainability (environment, economy and society), to better capture the interdependencies between the two concepts. The third proposition (P3) states that agility represents the strategic dimension of resilience, that is, the responsiveness of an organization to change. The fourth proposition (P4) points to digitization as an enabler for agility because technologies speed up processes by making the organization more flexible, efficient and effective. The fifth proposition (P5) notes that digitization, by promoting more efficient use of resources, has a positive impact on sustainability. The sixth preposition (P6), on the other hand, emphasizes that digitization, by triggering change can also exert a negative rebound effect on both resilience and sustainability. Finally, the seventh proposition (P7) concludes that a strategically resilient organization leverages its capabilities to deal with change, while also strengthening them.

The conceptual model shown in Figure 2 provides an overview on the connections among the topics discussed in this research, thus showing the relationships between resilience, sustainability and digitalization. The seven propositions introduced above will represent the vectors used to connect the topics previously discussed. The model clearly shows the interdependence existing between sustainability and resilience (P1), and it is this correlation that justifies a multidimensional vision of resilience (P2) exactly as it occurs for sustainability that is articulated in the three pillars of environment, economy and society. In an organizational perspective, the multi-attribute gives resilience a strategic implication that is enabled by agility (P3), that is, the ability of organizations to (re)act on change flexibly and responsively. Conversely, digital technologies can play both a positive (P5) and negative (P6) role on both sustainability and resilience, all depending on whether or not organizations are able to leverage these attributes to grow through a virtuous cycle (P7).

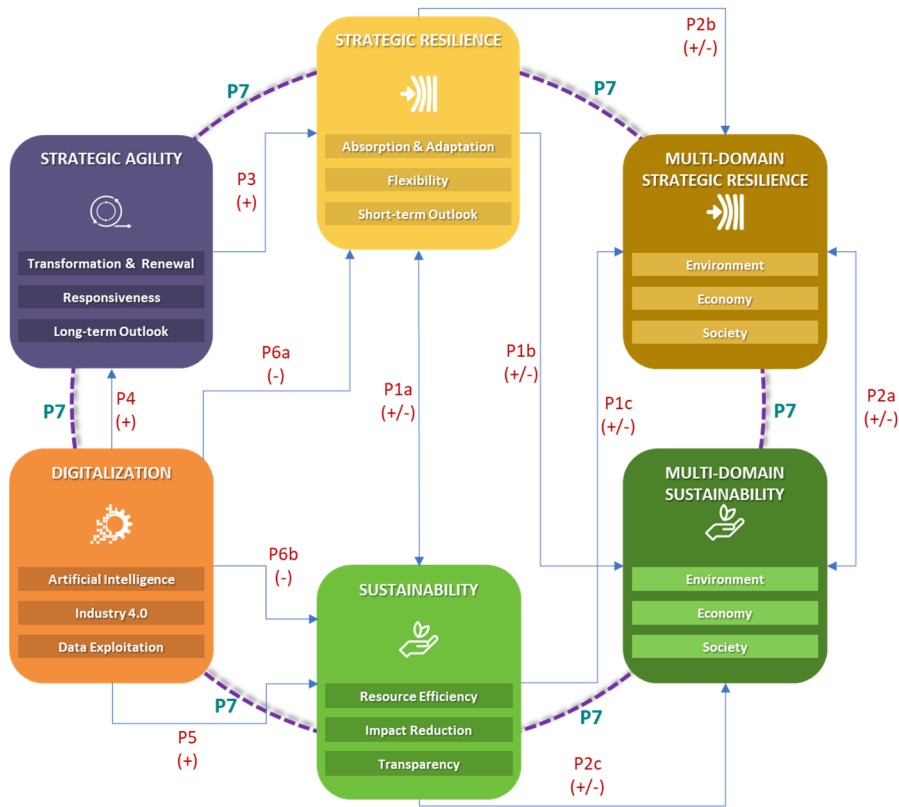

**Figure 2.** Conceptual model (own elaboration).

## 5. Discussion and Contribution

We set out to build, through a critical literature review, an explorative model to integrate resilience and sustainability. Through the combination of "isolated" areas, i.e., resilience work in management and sustainability literature, we identify and discuss key en-

ablers and model a route to build strategic multi-domain resilience, which is hypothesized as a pillar of an organization's sustainability. In so doing, we account for the complexities and interdependencies between environment, social and economic sub-systems of the firm and show that resilience needs to be considered holistically, similar to sustainability, in order to understand its overall performance implications better. Systems may be resilient but unsustainable as [69] with the resilient but unsustainable dictatorship illustrate. Thus, the use of resilience concepts for organizational decision making requires the clarification of the resilience of "what" and the addition of performance measures. Along the same lines of thought, we propose that resilience work would benefit from a normative stance with regard to the desirability of initial or new system conditions.

Additionally, dominant conceptualizations such as "absorption" or "adaptation capability" fall short in providing a proactive, opportunity-focused view of change or simply neglect disturbance that exceeds thresholds for absorption or adaptation [108]. Our model provides a way forward to thrive more than to survive under conditions of change and uncertainty. It is not only the resilience "of what" we mentioned above, also the resilience "to what" needs attention. In other words, it is crucial to determine better the forces of change that are at play. Change comes with many facets—from discontinuous to incremental, from reversible to irreversible, etc.—which will influence the firm's response and organization in order to build resilience. Large-scale disturbances require quite different organizational response mechanisms compared with minor disturbances, or gradual change [72], and they usually exceed thresholds for adaptation [109].

However, when focusing on agile business processes, flexibility and responsiveness become an integral part of the firm's structure and strategy and, thus, should prepare it well for an effective and resource-conserving answer to expected change and surprises of all types. More than being a planned response or process, strategic agility is a way of operating that accommodates changes. Agility can solve the tension between "preparedness" and efficiency that extant work has evidenced in the discussion on building redundancy and resourcefulness. It allows for process and resource flexibility and responsiveness at the same time, while, at the same time, being a resource-conservative way of operating. Agility adds time considerations, a crucial element in (re)acting to change. Furthermore, and importantly, it also helps to build long-term or strategic resilience, as it goes beyond absorption and adaptation to include transformation and renewal of the organization.

As we show, digitalization enables strategically agile processes. Digitalization, for example, with big data analysis can help predict change, and due to its unprecedented interconnectivity, it can facilitate communication and coordination with various stakeholders. At the same time, digitalization, by itself, may have positive or negative effects on both sustainability and resilience. Big data while being beneficial in terms of early detection of change may have a negative impact with regard to privacy issues and, thus, be in conflict with social sustainability. Digitalization itself, through its production and logistics, also comes with a high carbon footprint [110], which may trade off a positive impact due to less waste in production.

## 6. Conclusions and Future Research Directions

Although exploratory, our model illustrates the complexities and interactions that organizations face when dealing with resilience and sustainability on firm level. The framework helps in understanding the dynamics and the interplay between the concepts, cascading effects, potential trade-offs and synergies. The multi-domain view that we propose makes interdependencies clear and transparent and can inform prioritization. With more work and with empirical tests of our framework and propositions, other elements will arise, and trade-offs and synergies will become clearer. We also present a viable option on how to build strategic resilience effectively and efficiently and, in turn, sustainability in organizations. Designing and implementing strategically agile processes is one proposal on how to create strategic resilience, the integration of research insights from others, however, will yield many more that may inform organizations. As mentioned above, strategically

agile processes potentially allow us to solve the dilemma of efficiency versus efficacy in the organization's response to change. This comes close to the discussion of sets of capabilities, resources and structures that are necessary to create ambidextrous organizations, a body of research that is promising to further investigate the options for building strategic resilience. Additionally, literature on organizational adaptation presents a variety of approaches based on resource bases and capabilities [111] that describe how adaptation between the organization and the changing environment can be achieved, maintained or restored [112]. Another promising field for cross-fertilization is the research on business longevity or continuity, and the necessary ingredients, e.g., adaptability, flexibility, innovation [113,114] to it.

Even though our model accounts for interdependencies on the firm level, we neglect the interaction with the firm's environment, which definitely would merit close attention. Communities, networks and, more in general, ecosystems have been described to influence an organization's resilience. Setting boundaries, therefore, on the one hand helps us to manage complexity but on the other hand brings the disadvantage of omitting important relations. Future research should examine these interdependencies further. We have also assumed that sustainability, holistically, is an overarching objective of the organization, an assumption that may prove to be too optimistic. Additionally, in this context, internal versus external pressures and their interplay are promising areas to contribute to knowledge on resilience and sustainability of organizations.

Notwithstanding these limitations, we make a theoretical contribution through our framework and the strategic, multi-domain conceptualization of resilience. Additionally, we propose a novel approach to building strategic resilience via agile business processes, which presents a solution to tensions between efficiency and effectiveness of company response and strategy—a problem related to the current discussion on the creation of organizational resilience—and can be leveraged at the organizational level. Overall, we hope that the framework and the propositions help us to advance a step further towards resilience and truly sustainable organizations.

**Author Contributions:** Investigation, A.M and B.H.; conceptualization, A.M and B.H.; methodology, D.S.-B.; validation, B.H.; formal analysis, F.S.; data curation, F.S.; writing—original draft preparation, D.S.-B.; supervision, M.P.R.; project administration, M.P.R. All authors have read and agreed to the published version of the manuscript.

**Funding:** This research was co-funded by the Italian Ministry of Economic Development (D.M. 5 March 2018—CHAPTER II—Call for Research and Development Projects within the application areas consistent with the National Strategy of Intelligent Specialization [SNSI—Smart Factory), under the Project I.E.S.MAN. (Internet of Enterprise Sustainable Manufacturing) n. 211.

**Institutional Review Board Statement:** Not applicable.

**Informed Consent Statement:** Not applicable.

**Data Availability Statement:** Not applicable.

**Conflicts of Interest:** The authors declare no conflict of interest.

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
