# Peer review of "Thriving, Not Just Surviving in Changing Times: How Sustainability, Agility and Digitalization Intertwine with Organizational Resilience"

_sustainability, doi:10.3390/su13042052_

Round 1
Reviewer 1 Report
- The abstract doesn't provide the needed info on the scope of the paper, the methods, the results, and the conclusions. It should be rewritten following the suggested journal's format: 1) Background: Place the question addressed in a broad context and highlight the purpose of the study; 2) Methods: Describe briefly the main methods or treatments applied. 3) Results: Summarize the article's main findings; and 4) Conclusion: Indicate the main conclusions or interpretations.
- The introduction section could provide more information about the research gap the paper fills, explaining and mentioning other previous researches in the field and backgrounds. The authors can add more justification.
- There are some phrases that are repeated literally in the text (introduction and methodology): “i.e. proactive transformation and renewal and, in turn, sustainable firm performance”/ “review assumptions, formulate testable propositions, and finally, develop a conceptual framework. We conclude with the contribution and future research directions.” Please review and change them.
- Sustainability definition is poorly, and it must be extended regarding to its origin.
- Check spelling mistakes. (i.e. line 177: “whta” to “what”).
- Increasing the literature review support of 'The link between resilience and Sustainability' to improve understanding of Propositions 1 and 2.
- The authors must check all references carefully, there are some references that are cited in the text and vice versa (i.e. line 288).
Author Response
Dear Reviewer,
We are very grateful to you for pointing out the gaps in our manuscript. Your feedback guided us and helped us make significant improvements. Below are the changes and revisions we made.
Response to point 1
We have rewritten the abstract following the journal guidelines you kindly suggested.
Response to point 2
The introduction has also been totally revised, highlighting the gaps in the literature that our study aimed to fill. Argumentative justifications and links to other recently published studies have been provided for each gap.
Response to point 3
We apologize for the redundant repetitions that you kindly pointed out. These were misprints that we missed in the first version of the manuscript and have now corrected.
Response to point 4
Thank you for your suggestion. We have supplemented the section with a paragraph that introduces the concept of sustainability and defines it more clearly for the aims of this study.
Response to point 5
We checked and fixed spelling errors.
Response to point 6
In parallel with the integration of the sustainability paragraph, we have also introduced several more recent literature references that consolidate the statement of Prepositions 1 and 2.
Response to point 7
We cross-checked to resolve citation errors in the body of the text and in the list of bibliographic references.
We hope that we have fully responded to your suggestions.
Best regards, The Authors.
Reviewer 2 Report
The paper shows a good standard, well articulated and argued: the conclusions are clear and closely related. Nevertheless, more revisions are required due to the repetition of lines 65-79 of Paragraph " 1. Introduction" to lines 83-96 of Paragraph "2. Methodology and purpose of the research". This aspect causes a lack of clarity about the methodology adopted.
Author Response
Dear Reviewer,
Thank you very much for your kind words of appreciation to our research. We apologize for the misprint that we have resolved by eliminating the repetitions. We have also reworded the "Methodology" section to make it clearer to read as you suggested.
We hope that we have fully responded to your suggestions.
Best regards, The Authors.
Reviewer 3 Report
Using a critical literature review, the authors aimed to identify enablers and clarify the relationships between sustainability and resilience. Besides being very up-to-date thematic, this work is also a very ambitious one, as it also proposes a framework that explores ways to build resilience in the organization.
The work is exploratory and presents a model that illustrates the complexities and interactions that organizations face when dealing with resilience and sustainability on the firm-level. Although it lacks empirical tests, the work makes a good theoretical contribution to the relationships between resilience and sustainable organizations.
Author Response
Dear Reviewer,
Thank you very much for your kind words of appreciation to our research. For us it is a great support to continue the research that aims to achieve the empirical validation of the theoretical model we have proposed.
Best regards, The Authors.
Round 2
Reviewer 2 Report
The paper has been reviewed as requested, improving its structure, clarifying some issue and eliminating repetitions as in the old version.